# Hierarchical effects of historical and environmental factors on lizard assemblages in the upper Madeira River, Brazilian Amazonia

**Gabriela Marques Peixoto**[1ᴼ]*, **Rafael De Fraga**[2ᴼ], **Maria C. Araújo**[1], **Igor Luis Kaefer**[1,3ᴼ], **Albertina Pimentel Lima**[1ᴼ]

**1** Pós-Graduação em Ecologia, Instituto Nacional de Pesquisas da Amazônia, Manaus, Amazonas, Brasil, **2** Pós-Graduação em Recursos Naturais Amazônicos, Universidade Federal do Oeste do Pará, Santarém, Pará, Brasil, **3** Instituto de Ciências Biológicas, Universidade Federal do Amazonas, Manaus, Amazonas, Brasil

ᴼ These authors contributed equally to this work.
* gabriela.marquespd@gmail.com

**Data Availability Statement:** All relevant data are within the manuscript and its Supporting Information files.

**Funding:** Data collection was financially supported by Programa de Conservação da Vida Selvagem da

## Abstract

Investigating the role of historical and ecological factors structuring assemblages is relevant to understand mechanisms and processes affecting biodiversity across heterogeneous habitats. Considering that community assembly often involves scale-dependent processes, different spatial scales may reveal distinct factors structuring assemblages. In this study we use arboreal and leaf-litter lizard abundance data from 83 plots to investigate assemblage spatial structure at two distinct scales in southwestern Brazilian Amazonia. At a regional scale, we test the general hypothesis that the Madeira River acts as a barrier to dispersal of some lizard species, which results in distinct assemblages between river banks. At a local scale, we test the hypothesis that assemblages are not evenly distributed across heterogeneous habitats but respond to a continuum of inadequate-to-optimal portions of environmental predictors. Our results show that regional lizard assemblages are structured by the upper Madeira River acting as a regional barrier to 29.62% of the species sampled. This finding suggests species have been historically isolated at one of the river banks, or that distinct geomorphological features influence species occurrence at each river bank. At a local scale, different sets of environmental predictors affected assemblage composition between river banks or even along a river bank. These findings indicate that environmental filtering is a major cause of lizard assemblage spatial structure in the upper Madeira River, but predictor variables cannot be generalized over the extensive (nearly 500 km) study area. Based on a single study system we demonstrate that lizard assemblages along the forests near the banks of the upper Madeira River are not randomly structured but respond to multiple factors acting at different and hierarchical spatial scales.

Santo Antônio Energia S.A.; Conselho Nacional de Desenvolvimento Científico e Tecnológico (CNPq) and Coordenação de Aperfeiçoamento de Pessoal de Nível Superior (CAPES) granted PhD scholarships to GMP. CAPES provided a PNPD postdoc grant to RF and CNPq provided productivity grants to ILK and APL. The funders had no role in study design, data collection and analysis, decision to publish, or preparation of the manuscript.

**Competing interests:** The authors have declared that no competing interests exist.

## Introduction

Investigating historical and ecological factors structuring assemblages may reveal patterns of biodiversity distribution across time and space [1,2]. However, defining mechanisms and processes that potentially affect assemblage structure is often highly dependent on the spatial scale applied [3–5]. Such dependence results from the fact that assemblage composition (e.g. taxonomic diversity) is influenced by complex hierarchical interactions among processes that operate at multiple spatio-temporal scales [6]. In highly heterogeneous habitats such as the Amazonian tropical rainforests the relative contribution of historical and ecological processes to assemblage structuring is poorly understood for many taxa, mainly because multi-scale ecological approaches depend on standardized sampling systems, which have been specifically designed for such purpose [e.g. 7–12]. Regarding lizards, poor knowledge on assemblage structure also results from lack of refined data on individual species distribution [13], despite few unpublished studies have shown assemblage spatial structure defined by environmental heterogeneity [e.g. 14–16].

At broad spatial scales (e.g. Amazon Basin), it has been suggested that many organisms are restrictedly distributed by their inability to cross large rivers. From the classic studies of Alfred R. Wallace on primate distribution across the Amazon Basin [e.g. 17], it has been known that the Amazon River and some of its main tributaries (e.g. Madeira, Negro) may be important biogeographic barriers to dispersal. Testing the Wallace´s hypothesis has revealed the riverine barrier as a major factor explaining limited distribution of plants, frogs, birds, spiny rats, and primates [18–26]. Additionally, studies have shown that gene flow reduced or blocked by a riverine barrier may cause genotypic and phenotypic divergence in Amazonia [27–29]. Specifically for lizards, riverine barriers may cause intraspecific genetic divergence [27], although they do not necessarily produce different morphotypes [30]. Interspecifically, species distribution regionally limited to a single river bank may cause distinct assemblage compositions between banks [12,22,31].

At local scales, environmental predictors may affect species occurrence and abundance due to the filter effect of the spatial variation in habitat suitability [32,33]. In general, it is expected that habitat-specialist species find inadequate-to-optimum continuums of environmental conditions for survival and reproduction [34]. Environmental filtering has been found in Amazonia for plants, frogs, lizards, snakes, and birds [15,35–43]. For lizards, local assemblages may differ due to variation in individual abundance or species turnover along gradients of distance from water courses [31,44], elevation [45], climate seasonality [46], and number of trees [43,47]. Additionally, lizard assemblages may be indirectly structured by species turnover along gradients of canopy openness affecting the availability of thermoregulation sites [48,49], understory-plant density affecting the availability of foraging sites for perching species [50], and clay content in the soil affecting plant composition and food availability [43].

Integrating multiple spatial scales is relevant to estimating simultaneous effects of historical and ecological factors on assemblage structure, especially in heterogeneous habitats such as rainforests in Amazonia [51]. However, designing a sampling system which is efficient to quantify assemblages and habitats at multiple scales may be challenging. The RAPELD [1] method (Brazilian acronym for rapid sampling plus long-term ecological research) has been shown to be efficient for this purpose in the region of the upper Madeira River [13], due to (i) the adequate distribution of plot sets (5 km$^2$ each) so that hypotheses based on the effects of historical factors on regional assemblages may be tested (e.g. riverine barriers), and (ii) the plots following altitudinal contours reduce within-plot environmental variation, which allows them to be assumed as environmental units to test hypotheses based on environmental filtering [1]. The rationale behind testing such hypotheses in southwestern Amazonia is that the

Madeira River has been recognized as a barrier to dispersal of Squamata reptiles, which causes species turnover along a longitudinal gradient [52], and the region covers two endemism zones (Rondônia and Inambari) that are distinct regarding geological history and environmental heterogeneity [53].

In this study we use plot-based lizard abundance data from the upper Madeira River (southwestern Brazilian Amazonia) to investigate patterns of assemblage structure at two distinct spatial scales. At a regional scale, we test the hypothesis that lizard assemblages differ between the river banks. We expect differences in species composition and abundance as a consequence of the Madeira River historically limiting lizard dispersal. At local scale, we test the hypothesis that environmental heterogeneity causes species turnover, because species are absent or occur at low densities in suboptimal portions of environmental predictors. Specifically, we quantify the filtering effects on lizard abundance driven by gradients of number of trees, soil nutrient composition, shrub density, elevation, clay and sand content in the soil, and distance from the river bank. We expect that analyzing assemblages from two distinct perspectives will provide us with deep insights into factors that cause and maintain biodiversity at megadiverse regions such as the upper Madeira River basin.

## Materials and methods

### Study area

The study area is located near the banks of the upper Madeira River (coordinates of the centroid 08 ° 48004.0"S; 63 ° 56059.8"W), an important tributary of the Amazon River classified as a white and muddy river with a total length of 1,459 km. The upper Madeira River extends from the outskirts of Porto Velho (state of Rondônia) to about 600 km upstream, in southwestern Brazilian Amazonia, and its width varies from about 0.5 to 10 km depending on the river flow. The Madeira separates the Inambari and Rondônia endemism zones located along its left (west) and right (east) margins, respectively [20]. We also sampled plots close to the Jaci-Paraná River, a tributary on the east bank of the upper Madeira River (Fig 1).

In this study we quantified environmental heterogeneity as continuous gradients that may be broadly classified for descriptive purposes in three main habitat types. They mainly differ in canopy height, soil texture, understory-plant density, and species composition [following 54]. In the upland (terra-firme) forests habitats are never flooded by overflowing large rivers, the canopy is 30 m high, and the understory-plant density and clay content in the soil often depends on elevation [55]. The várzea forests are seasonally flooded by overflowing sediment-rich rivers, which produces nutrient-rich soils that are water-saturated for long periods. The canopy is 20 m high, and the understory is rich in bromeliads. The campinaranas are patches of palm tree-rich forests growing on a white-sand soil, which is highly drained and nutrient-poor [54].

The climate of the study area is tropical humid, with annual average temperature at 25.5 ˚C and average precipitation at 2,287 mm. Precipitation is distributed throughout the year in well-marked dry (May to September) and rainy (October to April) seasons. During the dry season, small streams can dry completely [56].

**Sampling design.** We collected arboreal and leaf-litter lizard abundance data in seven 5 $km^2$ RAPELD sampling sites (hereinafter modules), that were installed perpendicularly to the river bank. RAPELD [1] is a modification of the Gentry´s sampling method based on 1-ha plots [57], with the main difference being that the RAPELD plot central lines follow the altitudinal curves to reduce environmental variation within plots (PPBio—http://ppbio.inpa.gov. br). We sampled three modules on the east bank of the Madeira River (East Jirau, Jaci-Paraná and Morrinhos), and four modules on the west bank (West Jirau, Ilha das Pedras, Ilha dos

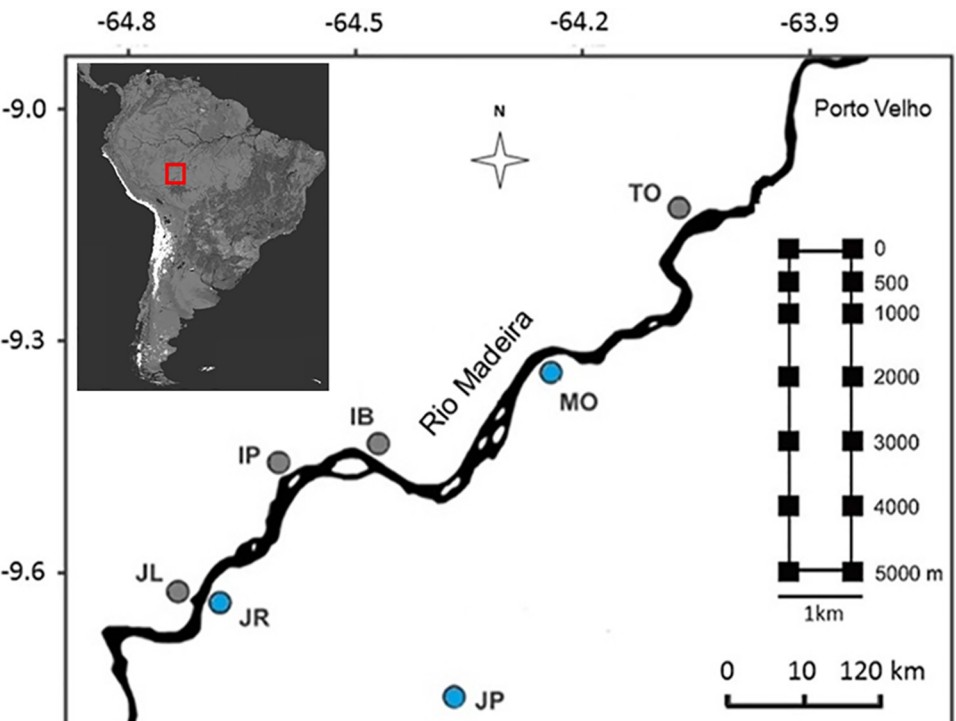

**Fig 1. Location of the upper Madeira River, state of Rondônia, Brazil.** Sampling 5 km² modules (circles) near the banks. Gray circles show modules in the Inambari endemism zone, blue circles are modules in the Rondônia endemism zone. The acronyms summarize sampling modules´ local names: TO = Teotônio, IB = Ilha dos Búfalos, IP = Ilha das Pedras, JL = West Jirau, JR = East Jirau, JP = Jaci-Paraná, MO = Morrinhos. In detail on the right side, the standard configuration of each module, with 14 plots (squares), 250 m-long each, distributed along a gradient of distance from the river bank (0–5,000 m).

Búfalos and Teotônio). The average distance between neighboring modules was 120 km. Each RAPELD module was composed of two 5-km long parallel trails, separated by 1 km. We surveyed seven 250 m plots (20 m wide) on each trail, totaling 98 plots (14 plots in each of the seven modules). The plots were distributed along a gradient of distance from the river bank, at 0, 500, 1000, 2000, 3000, 4000, and 5000 m.

We were not able to find lizards in 15 plots, and the excess of zeros in the dataset prevented us to reliably estimate pairwise distances among plots to summarize assemblage composition (see Data analysis). Therefore, we excluded zero-valued plots and our analyzes are based on 83 plots.

## Sampling effort and ethics

We sampled each plot in four different periods (24 February to 26 April 2010, 30 July to 19 August 2010, 5 November to 26 November 2010, and 13 January to 4 February 2011) to cover large portions of the regional variation in temperature and precipitation along a year. We used species' maximum abundance values per plot in the analyzes.

We found lizards using active visual search, with two simultaneous observers positioned 10 m apart. In addition, we supplemented the sampling effort by sweeping the leaf litter and removing debris in a 2 m strip following the center line of the plot. This approach was particularly useful to increase the efficiency of sampling leaf-litter species (e.g. Alopoglossidae, Gymnophthalmidae). Search on the vegetation and on the leaf litter was systematically conducted

in the first and second half of each visual sampling in the plots, respectively, thus constituting two different and complementary methods. The searching time in each plot varied between 40 and 60 minutes and was always conducted during the day.

We collected data under RAN-ICMBio / IBAMA permit 13777–2. IBAMA and ICMBio are institutes of Ministry of Environment, Government of Brazil. This permit was subject to approval of all procedures for collecting lizard abundance data.

**Environmental variables.** We measured eight environmental predictors in each plot in order to quantify spatial heterogeneity in habitat suitability. We quantified vegetation structure by measuring (i) number of trees and (ii) shrub density. Those predictors potentially affect abundance of tropical reptiles by influencing availability of foraging, resting, and thermoregulation sites [58–60]. We also measured edaphic gradients related to soil texture, fertility, and flat-level deviation, which are (iii) clay content, (iv) sand content, (v) nutrient composition (soil pH, Calcium, Magnesium, Potassium, Zinc, and exchangeable Aluminum), (vi) elevation, and (vii) terrain declivity. Those variables potentially affect lizard abundance by causing variation in the overall primary production [61] and availability of invertebrate prey [62]. Additionally, we measured (viii) distance from the river bank, because it has been found as a major factor structuring plant [36] and animal [31,38,39,41] assemblages in Amazonia. The methods used to measure each predictor are described in detail in Appendix 1.

## Data analysis

To quantify assemblage composition, we applied the Bray-Curtis index to estimate pairwise distances in species abundance among plots. We reduced dimensionalities using Principal Coordinate Analysis (PCoA) and represented assemblage composition by the first one or two axes produced (see below).

At regional scale (riverine barrier effects) we modeled the PCoA using all data (83 plots). The two first axes captured 30% (PCoA 1 = 16%. PCoA 2 = 14%) of the original variance in species abundance, and we used them to represent assemblage composition. To assess assemblage structuring, we used Multivariate Analysis of Variance MANOVA to test differences in assemblage composition (PCoA axes 1 and 2) between the river banks. We implemented a MANOVA using the vegan [63] R-package [64].

Analyzes at regional scale revealed two distinct lizard assemblages between the river banks (see Results). In addition, preliminary analyzes at local scale revealed that in two modules (Ilha das Pedras and East Jirau) environmental predictors may affect assemblage composition in opposite directions compared to the other modules (S1 and S2 Figs). These findings suggested that the banks of the Madeira River and some of the sampling modules along the same river bank are distinct environmental units, which contain distinct spatial structures of lizard assemblage composition. Therefore, to assess assemblage structure at local scale we modeled four distinct PCoA ordinations, using data from (i) the west bank, except for the module Ilha das Pedras (37 plots), which captured 86% of the original variance (PCoA 1 = 0.50, PCoA 2 = 0.36); (ii) the east bank, except for the module East Jirau (23 plots), which captured 45% of the original variance (PCoA 1 = 0.30, PCoA 2 = 0.15); (iii) the module Ilha das Pedras (12 plots), which captured 45% of the original variance (PCoA 1 = 0.32, PCoA 2 = 0.13); and (iv) the module East Jirau (11 plots), which captured 85% of the original variance (PCoA 1 = 0.49, PCoA 2 = 0.36).

The environmental predictors measured are expressed in different units and therefore in different orders of magnitude, so we transformed them using the "scale" function of the vegan R-package. This function subtracts mean values from each variable and scales centralized variables by dividing them by their standard deviation [63]. We used Mixed Linear Models to test

the effects of scaled environmental predictors on assemblage composition based on data from multiple sampling modules. By using this method, we were able to include sampling modules as random effects to minimize potential abrupt differences in environmental predictors and lizard assemblages among the modules analyzed in a same model [65]. We set up two different groups of mixed models, according to the assemblage compositions summarized by PCoA for the west and east banks of the Madeira River. Each group was composed of as many models as necessary to test all possible combinations of environmental predictors, except for those predictors that were highly correlated. For instance, clay and sand content in the soil were not used in a same model because they were highly correlated on both river banks (r ≥ 0.93). In addition, elevation was correlated with terrain declivity on both river banks (r ≥ 0.78) and soil-nutrient composition on the east bank (r = 0.66). The cut-off point was r = 0.51.

For the two modules that were analyzed separately (Ilha das Pedras and East Jirau), it was not necessary to control random effects of sampling sites, so we tested the effects of environmental predictors on the assemblage composition using multiple linear regression models. We tested models with assemblage composition (PCoA 1) as dependent variable, and all possible combinations of uncorrelated environmental predictors as independent variables.

To select the most parsimonious mixed-effects and multiple-regression models we ranked all the models by the corrected Akaike´s Information Criterion (AICc) [66]. We refined the model selection by penalizing nested models assuming ΔAICc < 2 as a cut-off point. All selected models were validated by normal distribution of residuals (Shapiro-Wilk W > 0.95, P > 0.05 in all cases).

For visually checking the distribution of lizard abundance values per species along river banks and environmental predictors (only those that significantly affected assemblage composition) we plotted ordinated sampling plots. These graphs will be used in this study for assessing how spread the distributions of abundance values are over the river banks and the environmental heterogeneity measured.

## Results

We found 27 lizard species, which are classified in 18 genera and 10 families. The most frequently found species were *Norops fuscoarautus* (Dactyloidae), *Gonatodes humeralis* (Sphaerodactylidae), and *Ameiva ameiva* (Teiidae) (Table 1), which occurred in both banks of the Madeira River, in 55, 49, and 30% of the plots respectively. Contrarily, *Alopoglossus angulatus* (Alopoglossidae) and *Enyalius leechii* (Leiosauridae) were found in one single plot.

### Regional assemblage structuring—Madeira River as a biogeographic barrier

We found 19 species on both banks of the Madeira River, which is equivalent to 70.37% of the total diversity sampled. This finding suggests that most of the species sampled are widely distributed throughout the study area. However, for several of the species found on both sides of the river (e.g. *Loxopholis percarinatum*, *Kentropyx altamazonica*, *Cercosaura eigenmanni*, *Plica plica*, *Uranoscodon superciliosus*, *Copeoglossum nigropunctatum*), plot-related frequency and abundance were not even between the river banks (Fig 2). Additionally, five species (18.52%) were restricted to the west bank–*Alopoglossus angulatus*, *Norops tandai*, *Dactyloa transversalis*, *Cercosaura bassleri*, and *Kentropyx pelviceps*, and three species (11.11%) were restricted to the east bank–*Arthrosaura reticulata*, *Kentropyx calcarata*, and *Enyalius leechii*. These findings suggest two distinct assemblage compositions delimited by the Madeira River, which is strongly supported by differences in the PCoA scores (based on 83 plots) between the river banks (MANOVA Pillai Trace = 0.315, $F_{1-81}$ = 18.40, P <0.001).

**Table 1. List of lizard species sampled in the upper Madeira River, Brazil.** N = total abundance per species, East and West = Madeira River banks filled with presence (1) and absence (0) data.

| Family/Species | N | East | West |
|---|---|---|---|
| **Dactyloidae** | | | |
| Norops fuscoauratus (D'Orbigny, 1847) | 103 | 1 | 1 |
| Norops tandai (Wagler, 1830) | 2 | 0 | 1 |
| Norops ortonii (Cope, 1869) | 2 | 1 | 1 |
| Dactyloa punctata (Daudin, 1802) | 27 | 1 | 1 |
| Dactyloa transversalis (Dumeril, 1851) | 9 | 0 | 1 |
| **Alopoglossidae** | | | |
| Alopoglossus angulatus (Linnaeus, 1758) | 2 | 0 | 1 |
| **Gymnophthalmidae** | | | |
| Arthrosaura reticulata (O'Shaughnessy, 1881) | 5 | 1 | 0 |
| Cercosaura argulus (Peters, 1863) | 5 | 1 | 1 |
| Cercosaura eigenmanni (Griffin, 1917) | 11 | 1 | 1 |
| Cercosaura bassleri (Ruibal, 1952) | 8 | 0 | 1 |
| Iphisa elegans (Gray, 1851) | 8 | 1 | 1 |
| Loxopholis percarinatum (Muller, 1923) | 10 | 1 | 1 |
| **Hoplocercidae** | | | |
| Enyalioides laticeps (Guichenot, 1855) | 3 | 1 | 1 |
| Hoplocercus spinosus (Fitzinger, 1843) | 2 | 1 | 1 |
| **Leiosauridae** | | | |
| Enyalius leechii (Boulenger,1885) | 2 | 1 | 0 |
| **Scincidae** | | | |
| Copeoglossum nigropunctatum (Spix, 1825) | 14 | 1 | 1 |
| **Phyllodactylidae** | | | |
| Thecadactylus rapicauda (Houttuyn, 1782) | 21 | 1 | 1 |
| **Sphaerodactylidae** | | | |
| Chatogekko amazonicus (Andersson, 1918) | 12 | 1 | 1 |
| Gonatodes hasemani (Griffin, 1917) | 29 | 1 | 1 |
| Gonatodes humeralis (Guichenot, 1855) | 432 | 1 | 1 |
| **Teiidae** | | | |
| Kentropyx altamazonica (Cope, 1876) | 14 | 1 | 1 |
| Kentropyx calcarata (Spix, 1825) | 37 | 1 | 0 |
| Kentropyx pelviceps (Cope, 1868) | 29 | 0 | 1 |
| Ameiva ameiva (Linnaeus, 1758) | 48 | 1 | 1 |
| **Tropiduridae** | | | |
| Plica plica (Linnaeus, 1758) | 7 | 1 | 1 |
| Plica umbra ochrocollaris (Spix, 1825) | 21 | 1 | 1 |
| Uranoscodon superciliosus (Linnaeus, 1758) | 5 | 1 | 1 |
| **Total** | **868** | | |

## Local assemblage structuring—The role of environmental predictors

On the west bank of the Madeira River (except for the module Ilha das Pedras) three mixed-effects models were selected by ΔAICc < 2 (Table 2). All the selected models consistently returned number of trees as a major gradient affecting assemblage composition (P < 0.001 in all cases). Despite some species occupied large portions of the gradient of number of trees (e.g. *Ameiva ameiva*, *Norops fuscoauratus*), species absence or low abundance in specific intervals between 144 and 613 trees caused species turnover (Fig 3). According to the models selected,

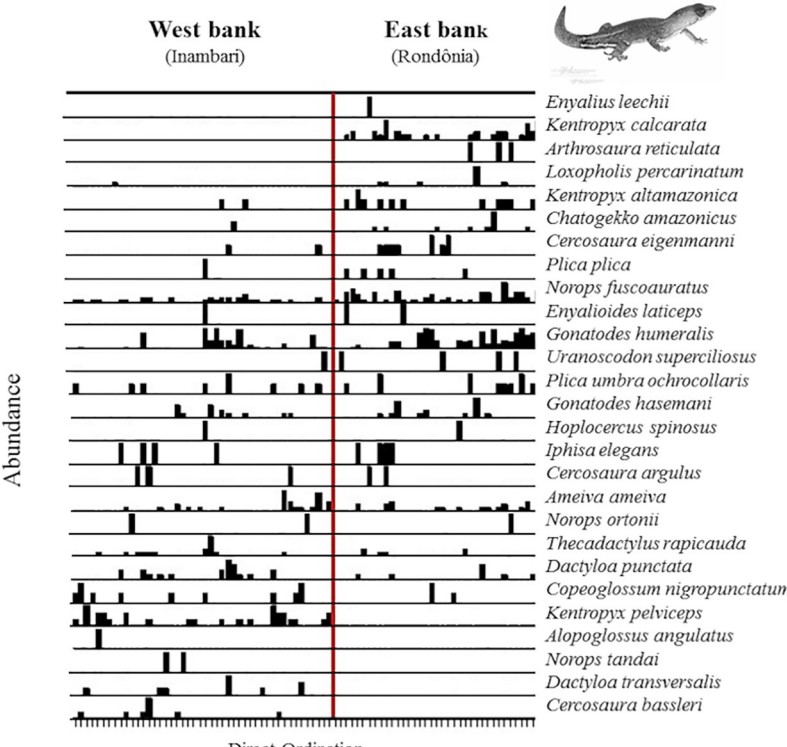

**Fig 2. Plots ordinated according to their position in the upper Madeira River (west or east bank).** The heights of the black rectangles are relative to species abundance values.

**Table 2. Summary of the results returned by linear mixed-effects models.** The models were set up using data from the west (Teotônio, Ilha dos Búfalos, and West Jirau) and east (Morrinhos and Jaci-Paraná) banks of the upper Madeira River. The models were selected by ΔAICc < 2. Shapiro-Wilk tests were applied on the residuals from each model to test normality. Bolded p-values show cases in which the null hypothesis was rejected.

| Margins | Fixed effects | AICc | Weight | df | t | p | Total variance | Shapiro-Wilk |
|---------|---------------|------|--------|-----|-----|-----|----------------|--------------|
| **West** | Number of Trees and Elevation | 12.89 | 0.314 | Intercept:2.18 | -6.92 | **<0.001** | 54% | P = 0.109 |
| | | | | Trees:3.01 | 18.8 | **<0.001** | | |
| | | | | Elevation:1.43 | -0.31 | 0.76 | | |
| | Sand and Number of Trees | 12.88 | 0.309 | Intercept:3.39 | -14.69 | **0.001** | 69% | P = 0.066 |
| | | | | Sand:1.86 | -0.19 | 0.84 | | |
| | | | | Trees:3.29 | 18.8 | **<0.001** | | |
| | Clay and Number of Trees | 12.88 | 0.305 | Intercept:1.00 | -9.07 | **<0.001** | 76% | P = 0.153 |
| | | | | Clay:9.76 | 0.10 | 0.91 | | |
| | | | | Trees:3.27 | 18.88 | **<0.001** | | |
| **East** | Elevation and Margin distance | 2.0 | 0.412 | Intercept:2.30 | 6.37 | **<0.001** | 71% | P = 0.782 |
| | | | | Elevation:2.30 | -6.27 | **<0.001** | | |
| | | | | Margin:2.30 | 1.72 | 0.09 | | |
| | Number of Trees and Elevation | 2.0 | 0.400 | Intercept:2.30 | 5.92 | **<0.001** | 72% | P = 0.413 |
| | | | | Trees:2.10 | 18.8 | 0.10 | | |
| | | | | Elevation:2.30 | -0.31 | **<0.001** | | |

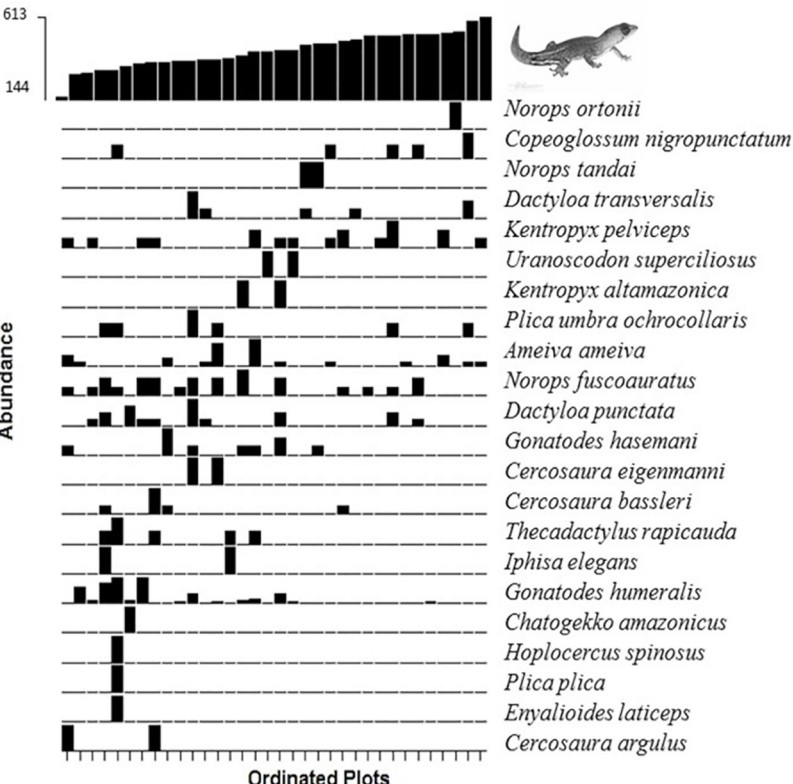

**Fig 3. Plots ordinated according to their position relative to the number of trees measured in the west bank of the upper Madeira River, state of Rondônia, Brazil.** The heights of the black rectangles depict the relative species abundances.

assemblage composition was not affected by elevation (P = 0.76), and soil-content of sand (P = 0.84) or clay (P = 0.91).

Three multiple-regression models were selected for Ilha das Pedras sampling module (Table 3), all of them containing elevation as an independent variable. This predictors significantly affected assemblage composition according to a model constructed with soil sand content as an additional independent variable (P = 0.05) (Fig 4). However, the effects of elevation on the assemblage composition were marginally significant in models containing number of trees (P = 0.06) and soil clay content (P = 0.07) as independent variables.

On the east river bank (except for the East Jirau module) two models were selected as most parsimonious. Both models consistently showed strong effects of elevation on assemblage composition (P < 0.001 in both cases). This finding suggests species turnover along an elevational gradient of 69.12–100.59 m (Fig 5). According to the same models, distance from the river bank (P = 0.09) and number of trees (P = 0.1) did not affect assemblage composition.

Two multiple-regression models were selected for the East Jirau module. Both models consistently returned distance from the river bank (Fig 6) as a relevant gradient affecting assemblage composition (P < 0.001 in both cases). Soil-content of sand (P = 0.24) and clay (P = 0.13) did not affect assemblage composition.

## Discussion

At regional scale, we found that lizard assemblages are spatially structured by differences in assemblage composition between river banks. This finding is consistent with large Amazonian

**Table 3. Summary of the results returned by linear mixed-effects models.** The models were set up using data from the Ilha das Pedras (west river bank) and East Jirau (east river bank) modules to test the effects of environmental predictors on lizard assemblage composition. The models were selected by ΔAICc < 2. Shapiro-Wilk tests were applied on the residuals from each model to test normality. Bolded p-values show cases in which the null hypothesis was rejected.

| Margins | Variables | AICc | Weight | Std. error | t | P | F | r2 |
|---|---|---|---|---|---|---|---|---|
| West | Number of Trees and Elevation | 15.7 | 0.31 | Intercept:8.76 | 0.00 | 1.00 | 2.746 | 0.37 |
| | | | | Trees:9.19 | 0.90 | 0.39 | | |
| | | | | Elevation:9.19 | -2.06 | 0.06 | | |
| | Sand and Elevation | 15.8 | 0.29 | Intercept:8.81 | 0.00 | 1.00 | 2.66 | 0.37 |
| | | | | Sand:9.21 | 0.85 | 0.42 | | |
| | | | | Elevation:9.21 | -2.18 | **0.05** | | |
| | Clay and Elevation | 16.2 | 0.25 | Intercept: 1.21 | 0.1 | 1.10 | 2.45 | 0.35 |
| | | | | Clay:6.12 | -0.64 | 0.53 | | |
| | | | | Elevation:1.88 | -1.98 | 0.07 | | |
| East | Clay and Distance from the margin | 6.5 | 0.655 | Intercept: 5.69 | 0.00 | 1.00 | 15.42 | 0.81 |
| | | | | Clay: -1.05 | -1.69 | 0.13 | | |
| | | | | Margin:2.89 | 4.63 | **<0.001** | | |
| | Sand and Distance from the margin | 7.9 | 0.367 | Intercept: 2.30 | 5.92 | **<0.001** | 13.07 | 0.78 |
| | | | | Sand:8.81 | 1.28 | 0.24 | | |
| | | | | Margin:2.85 | 4.15 | **<0.001** | | |

rivers acting as dispersal barriers for several organisms, which have caused different species subsets composed of plants [18], diurnal frogs [12], birds [19,21,22], and primates [24]. At local scale, we showed that lizard assemblages are spatially structured by species turnover along environmental predictors. However, a set of environmental predictors cannot be assumed as generalized predictors among sampling sites. Our overall results are broadly consistent with those obtained for frog assemblages sampled in the same plots [12], which suggests

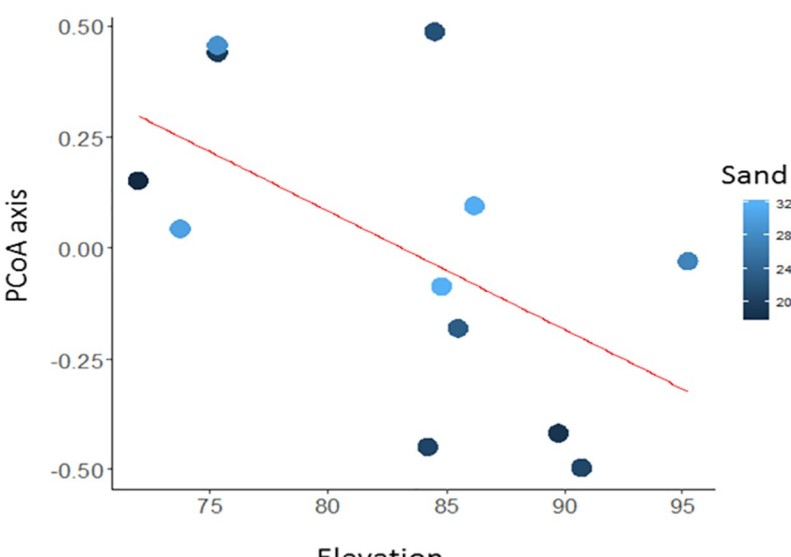

**Fig 4. Partials from a multiple linear model for the Ilha das Pedras module.** Model for the effects from the elevation and sand contents in the soil on lizard assemblage composition. Assemblage composition was summarized by the first axis of a Principal Coordinates Analysis based on abundance data of the upper Madeira River, state of Rondônia, Brazil. The shades of blue show values of sand content in the soil.

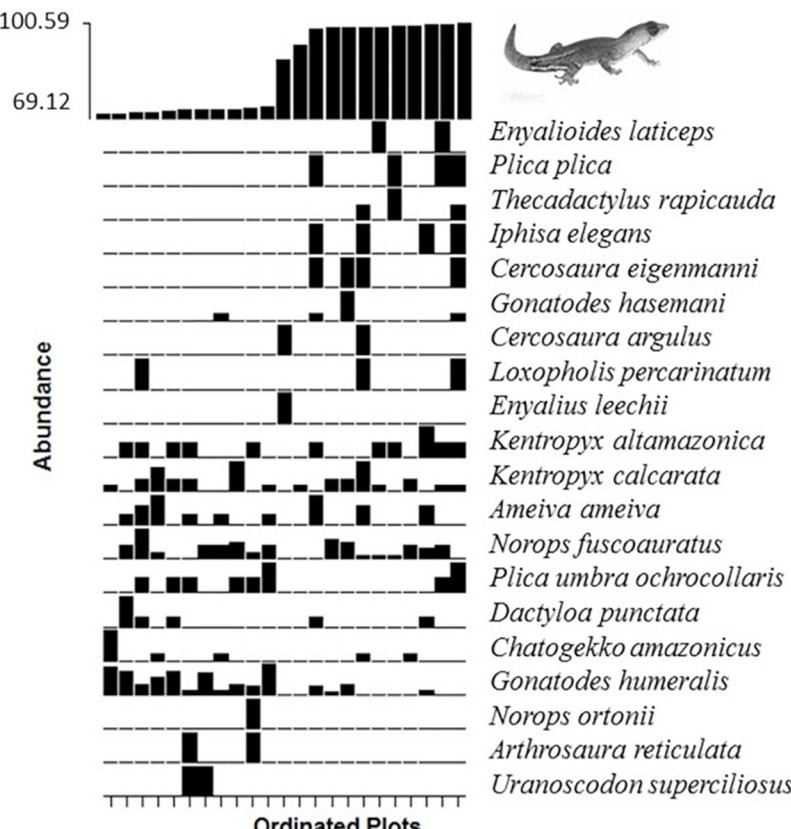

**Fig 5. Plots ordinated according to their position relative to a gradient of elevation (meters above the sea level) in the east bank of the upper Madeira River, state of Rondônia, Brazil.** The heights of the black rectangles depict the relative species abundances.

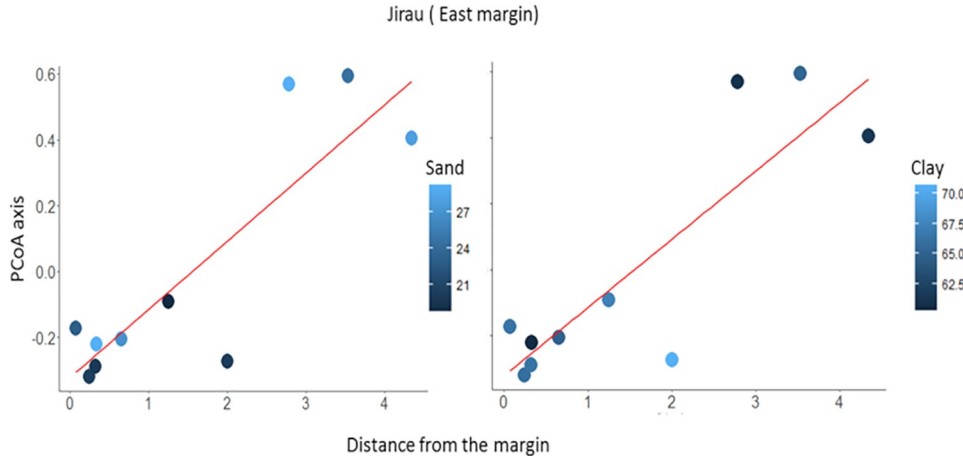

**Fig 6. Partials from a multiple linear model from the East Jirau module.** The effects of distance from the river bank, sand, and clay contents in the soil on lizard assemblage composition. Assemblage composition was summarized by the first axis of a Principal Coordinates Analysis based on abundance data from the East Jirau sampling module, located on the east bank of the upper Madeira River, state of Rondônia, Brazil. The shades of blue show values of sand and clay contents in the soil.

multi-taxa ecological patterns. We relied on a single dataset to provide understanding about assemblage structure based on interacting historical and ecological processes. Therefore, we highlight the relevance of investigating multi-scale assemblage structuring for ecology and conservation decision making.

In the upper Madeira River, assemblage divergence between river banks has been attributed to historical processes regionally reducing species dispersal [12], and delimiting the Amazonian endemism zones Inambari and Rondônia [20]. Approximately half of the species present in the assemblage of diurnal frogs (13 species) of that region were restricted to one of the river banks [12]. The smaller proportion of regionally isolated lizard species (29.63%) is reasonably explained by the lower dispersal capacity of small and site-attached frogs compared with most lizards. A taxonomic bias may be also contributing to this scenario, since there are consistent and recent efforts to investigate the taxonomic status of frog species in the region [12], and such efforts are unparalleled regarding lizards. Nonetheless, we investigated assemblages in which about 30% of the sampled species were isolated by the river, and another 30% of the species occurred at low relative frequency or abundance at one of the river banks. This was a sufficiently adequate scenario to assume the river as a historical factor segregating assemblages between the river banks. Even though the riverine-restricted geographic distribution of species such as *Kentropyx calcarata* observed in this study is supported by basin-level data, we highlight that most of regionally isolated species in our sample are widely distributed throughout Amazonia outside our study area [13]. Such inconsistency may be explained by the strength of the river as a dispersal barrier varying along the river course, or even being nullified in response to meandering shapes [30,67–69]. Additionally, the barriers may be seasonal, because bridges for stepping-stone dispersal may be revealed during the dry season, which allows gene flow between river banks [70]. Therefore, our results for assemblage structure at regional scale should not be extrapolated to unsampled stretches of the Madeira River or other Amazonian rivers, because lizards probably have found multiple dispersal routes through evolutionary time [27].

The isolation of species on one of the river banks may be related to the geomorphological heterogeneity of the Madeira River across our study area. The Madeira river flows over an incisive fluvial valley, with predominantly crystalline and a geologically ancient basement (ca. 16 Ma). The morphodynamical development was mainly influenced by the geomorphological and climatic changes resulting from the Andean Orogeny in the Cenozoic [71], which have produced a relatively stable course along recent geological times [72]. Such stability in the shape of the river course has prevented meandering across most of the study area, which could facilitate for species to cross the river [73]. Exceptionally, the modules located further upstream (East and West Jirau) have rocky outcrops that are exposed in the middle of the river course during the dry season, which can act as bridges for stepping-stone dispersal (field observation). Although lizard species used alternative dispersal routes to widespread their distribution throughout Amazonia, our study showed that they were regionally prevented from colonizing or maintaining populations on both banks of the upper Madeira River. One could argue that our results of a river-barrier effect are biased due to the low detection probability of lizards, which resulted in false absence of species [74,75]. However, we think that a possible sampling bias was mitigated by the large sampling effort associated to the combination of different visual sampling methods employed in this study.

Besides affecting assemblage composition, the effect of rivers as barriers can also be observed at the intraspecific level in different biogeographic domains, resulting in genetic and morphological divergence among lizard populations due to restriction to gene flow [30,76,77]. Studies on the genetic and phenotypic differentiation of populations of a same species on opposite banks of the Madeira River should be performed as they might help to understand the initial steps of allopatric speciation in Amazonian lizards.

At local scale, lizard assemblages were spatially structured by environmental filtering causing non-random assemblage composition. Environmental conditions selected species that were unable to survive and maintain viable conditions in given sampling plots [78]. Despite we sampled species that are generalist in relation to the environmental predictors measured (e.g. *Ameiva ameiva*, *Norops fuscoauratus*), species for which distributions were restricted to narrow regions of gradients (e.g. *Cercosaura argulus*, *Norops ortonii*, *Uranoscodon superciliosus*) caused species turnover across sampling plots. Species turnover mediated by environmental filtering is a major factor structuring local assemblages in Amazonia [e.g. 41,36,39], and in the upper Madeira River it has efficiently explained assemblage structure in frogs [12], snakes [79], and bats [80]. However, we cannot generalize a single environmental dataset as a predictor for assemblage composition in all plots. Environmental predictors for assemblage composition differed between the river banks or even along a same river bank. This finding suggests that the scale at which lizard assemblages respond to environmental heterogeneity may be more refined than the classification of the Madeira River banks as distinct endemism zones [20,81].

Number of trees was a major factor causing species turnover in the west bank of the Madeira River. This gradient ranged from 144 to 613 trees, which shows that the vegetation structure is quite heterogeneous throughout our study area. Heterogeneity in vegetation structure affects occurrence and abundance of tropical squamates due to variation in the availability of foraging, nesting, resting, and thermoregulating sites [58,60]. Additionally, tree cover may directly affect food availability, protection against predators, light intensity, temperature, humidity, and wind speed [59,60]. The evidence for assemblage structuring along a gradient of number of trees is of concern from a conservation point of view, because our study area has been intensely deforested by the agribusiness and large hydroelectric plants [82]. It is widely expected that species dependent on high levels of tree cover (e.g. *Norops tandai*, *Norops ortonii*, *Dactyloa transversalis*) will either be locally extinct or migrate to more suitable habitats.

We found species turnover along an elevational gradient, although this finding was most evident on the east bank of the Madeira River. On the east bank the plots were installed on the depression of the Ji-Paraná River, which generated elevation values below 30 m. Low elevation is often related to outcropping of groundwater and high drainage density [83,71], which favors the occurrence of habitat-specific species for high humidity. For instance, *Arthrosaura reticulata* and *Uranoscodon superciliosus* typically occupy humid low areas [84,85], and in this study those species were found only on the east bank of the Madeira River. Additionally, elevation indirectly influences assemblage composition because it affects water availability and soil fertility [86,87], and therefore the overall structure of available habitats [88,89]. Extreme variation in elevation may cause behavioral and morphological differentiation in lizards [90]. In this study we showed that even subtle variation in elevation (24 to 128 m) may be sufficient for species to be locally filtered. A similar finding was observed using frog assemblage data from the Guiana Shield [37].

The gradient of distance from the river caused species turnover in the East Jirau module. Although habitats may be classified in riparian and non-riparian zones [91], gradients of distance from water courses carry multiple continuous interacting variables of microclimate, nutrient availability, vegetation cover, and edaphic structure. Habitats continuously changing along gradients of distance from streams (< 12 m wide) have caused species turnover structuring plant [36], frog [38], snake [41], and bird [39] assemblages. We have shown a similar pattern using lizard abundance data, with the main difference being that the gradient we measured refers to the distance from the bank of one of the major tributaries of the Amazon River. However, no significant effect of distance from the river on assemblage composition was observed using data from the other modules. This finding suggests that assemblages diverging

between riparian and non-riparian zones should not be generalized in relation to gigantic rivers, or assemblage segregation should occur at distances that are greater than 5 km away from the river bank.

Some of the results found may be associated to environmental variables that were not explicitly measured in this study. For example, *Hoplocercus spinosus* (Hoplocercidae) occurred on both banks of the upper Madeira River but occurrence was restricted to plots with rocky outcrops. Such condition was only found in the westernmost sampling modules of the study area (East and West Jirau), where the species finds optimal availability of thermoregulation and refuge sites [91]. This finding reflects relationships between species and habitats that are dependent of biological traits affecting survival [92,93] and dispersal capacity [94,95], such as body size, diet [96], specificity level in habitat use [89], reproductive [49], and foraging mode [97]. Therefore, although patterns of assemblage structure are usually described based on dissimilarities among plots regarding subsets of cooccurring species, they may be determined by ecological requirements of individual species.

We have shown that lizard assemblages in the upper Madeira River are structured by scale-dependent hierarchical factors. Historical processes related to the Andes uplift [98] have isolated regional assemblages between the river banks, and have also generated distinct habitat patches, which in turn generate distinct local lizard assemblages. It is generally well established that interacting historical and environmental factors explain hierarchical structures of assemblages [5]. However, empirical application is not common because it relies on efficient sampling designs to capture multiple scales [1]. In the megadiverse Amazonian rainforests this has been achieved by a few studies [12,22,31,73]. Considering the fine levels in which those studies have understood processes affecting biodiversity, efficient methods for multi-scale sampling should be prioritized by ecology and conservation biology.

## Supporting information

**S1 Data. Protocols for measuring the environmental predictors.** Predictors used as independent variables in the ecological models to test lizard assemblage structuring in the upper Madeira River, Brazilian Amazonia.
(PDF)

**S1 Fig. Partials from Mixed Linear Models for the west bank of the Madeira River.** Effect of environmental predictors on lizard assemblages composition (PCoA axis 1). The models were selected by ΔAICc < 2. (A) Ilha das Pedras (B) Ilha dos Búfalos (C) West Jirau (D) Teotônio.
(PDF)

**S2 Fig. Partials from Mixed Linear Models for the east bank of the Madeira River.** Effect of environmental predictors on lizard assemblages composition (PCoA axis 1). The models were selected by ΔAICc < 2. (A) Jaci-Paraná (B) East Jirau (C) Morrinhos.
(PDF)

## Acknowledgments

We thank Stefan Lötters and an anonymous reviewer for insightful suggestions on this manuscript. Data collection was logistically supported by Programa de Pesquisas em Biodiversidade (PPBio), Centro de Estudos Integrados da Biodiversidade Amazônica (INCT-CENBAM), and Programa de Conservação da Vida Selvagem da Santo Antônio Energia S. A. Conselho Nacional de Desenvolvimento Científico e Tecnológico (CNPq) and Coordenação de Aperfeiçoamento de Pessoal de Nível Superior (CAPES) granted PhD scholarships

to GMP. CAPES provided a PNPD postdoc grant to RF and CNPq provided productivity grants to ILK and APL.

## Author Contributions

**Conceptualization:** Gabriela Marques Peixoto, Rafael De Fraga, Maria C. Araújo, Igor Luis Kaefer, Albertina Pimentel Lima.

**Data curation:** Gabriela Marques Peixoto, Maria C. Araújo, Albertina Pimentel Lima.

**Formal analysis:** Gabriela Marques Peixoto.

**Funding acquisition:** Albertina Pimentel Lima.

**Investigation:** Gabriela Marques Peixoto, Albertina Pimentel Lima.

**Methodology:** Gabriela Marques Peixoto, Rafael De Fraga, Albertina Pimentel Lima.

**Project administration:** Igor Luis Kaefer, Albertina Pimentel Lima.

**Supervision:** Igor Luis Kaefer, Albertina Pimentel Lima.

**Writing – original draft:** Gabriela Marques Peixoto, Rafael De Fraga.

**Writing – review & editing:** Gabriela Marques Peixoto, Rafael De Fraga, Igor Luis Kaefer, Albertina Pimentel Lima.

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
