## [Decision Letter · Decision Letter 0]

17 Jan 2020

PONE-D-19-15180

Hierarchical effects of historical and environmental factors on lizard assemblages in the upper Madeira river, Amazonian Brazil

PLOS ONE

Dear Mrs. Marques Peixoto,

Thank you for submitting your manuscript to PLOS ONE. After careful consideration, we feel that it has merit but does not fully meet PLOS ONE’s publication criteria as it currently stands. Therefore, we invite you to submit a revised version of the manuscript that addresses the points raised during the review process.

It was difficult to find referees and eventually only one replied.The review was very detailed, however, so that my decision "major revision" could well be justified. 

The referee suggests that the paper has potential but the referee warns that results could be misinterpreted. The referee's concern is that data are based on count and presence/absence data only and gives concrete literature hints. The solution proposed is the employment of modeling techniques that account for potential sampling errors. Moreover, the referee proposes to compare findings with results from other lizard systems around the world. For details, see below.

I strongly go along with the critics and the recommendations by the referee.

We would appreciate receiving your revised manuscript by Mar 02 2020 11:59PM. To enhance the reproducibility of your results, we recommend that if applicable you deposit your laboratory protocols in protocols.io, where a protocol can be assigned its own identifier (DOI) such that it can be cited independently in the future. For instructions see: http://journals.plos.org/plosone/s/submission-guidelines#loc-laboratory-protocols

We look forward to receiving your revised manuscript.

Kind regards,

Stefan Lötters

Academic Editor

PLOS ONE

Journal Requirements:

2. We note that Figure 1 in your submission contain [map/satellite] images which may be copyrighted. All PLOS content is published under the Creative Commons Attribution License (CC BY 4.0), which means that the manuscript, images, and Supporting Information files will be freely available online, and any third party is permitted to access, download, copy, distribute, and use these materials in any way, even commercially, with proper attribution. For these reasons, we cannot publish previously copyrighted maps or satellite images created using proprietary data, such as Google software (Google Maps, Street View, and Earth). For more information, see our copyright guidelines: http://journals.plos.org/plosone/s/licenses-and-copyright.

Additional Editor Comments (if provided):

Reviewers' comments:

Reviewer's Responses to Questions

**Comments to the Author**

1. Is the manuscript technically sound, and do the data support the conclusions?

Reviewer #1: Partly

2. Has the statistical analysis been performed appropriately and rigorously? 

Reviewer #1: No

3. Have the authors made all data underlying the findings in their manuscript fully available?

Reviewer #1: Yes

4. Is the manuscript presented in an intelligible fashion and written in standard English?

Reviewer #1: Yes

5. Review Comments to the Author

Reviewer #1: In this manuscript, authors sample lizard assemblages in 83 plots from southwestern Amazon, Brazil to investigate spatial structure in two distinct scales. They found differences in species presences and relative abundances, suggesting that the Madeira River acts as a barrier for almost 30% of the species. At the local scale, they found that different predictors were important for species.

The manuscript is interesting and includes good data and sampling. However, I am not sure about the conclusions, since inference was based on data that may suffer from sampling errors. Additionally, the conclusions on the barrier effect may be overstated given the sampling design and time frame of the study. Below I detailed some major and minor comments and suggestions, which I hope help authors in some way.

Major comments

My main concern is that all inference here is based on count and presence/absence data, which may be influenced by sampling errors. For instance, you may find 20 individuals of species A and 19 individuals of species B in a given plot, when in fact there was 120 individuals of species A and 20 individuals of species B. This is typically a false-negative type error. Alternatively, you may have misidentified species B, inducing false-positive errors in your counts. Similar problems occur when using presence/absence data when absence is not included and estimated in the model. This subject is well covered in the literature. You may want to have a look at Kéry and Schmidt 2008 (Community Ecology 9: 207-216), Fitzpatrick et al. 2009 (Ecological Applications 19: 1673-79), Miller et al. 2012 (Ecological Applications 22: 1665-74) and so on. Since sampling effort seems to be enough, I recommend applying a modeling technique that accounts for sampling errors (for instance, N-mixture models), which I find more appealing to discuss results arising from count data. However, I understand that you would have to make a major change in your manuscript, especially if results change, which I would not be surprised. At least, I recommend including a paragraph discussing this subject, since I find hard to believe that you have captured the exactly number of individuals and species present at a given site, and thus, the results may be biased.

Your sample includes temporal variation (extended for a year, or so) but there are no temporal covariates among your predictors. Lizard activity is usually related to temperature, sunlight, rainfall and so on. You probably sampled plots in different days and different hours within a day, and ciclicity occurs even within individuals. How do you think the absence of temporal predictors affect your results? For instance, Figure 2 shows species ‘abundances on each river bank. Some species seems to be common on one bank, but not the other. Is this really the barrier effect of the river? By ignoring temporal variability, you assume sampling did not vary along the year.

Authors suggest that the pattern found – the river acting as a barrier – was important. However, they mention that the lizard fauna found is widely distributed over the Amazon. This is somewhat confusing to me because if species were well-distributed, they would probably be found everywhere. To explain that, authors mention that the barrier effect of the river may vary, suggesting seasonality. This makes me wonder if species are truly isolated, as suggested. The barrier effect could be only temporarily and the results found are explained by the time frame of the study. I would like authors to clarify that point.

I think your manuscript would benefit from contrasting your results with other lizard systems around the world. For instance, are there examples of rivers acing as barriers for lizards elsewhere? What about other reptiles? What are the covariates related to isolation for lizards or reptiles in general? In addition, you could provide some information about the Amazon basin. For instance, what is the mean width of the big rivers? Finally, concerning the species found. What is the habit of the most abundant species? Did you find terrestrial, arboreal and fossorial species equally? Does you results relate to species ‘habits? I believe that finding terrestrial species (the teiids, for instance) may be easier than arboreal (Enyalius?) or fossorial. In the same way, diurnal species will be easier to find during the day and the opposite for nocturnal. How was the sampling scheme? Can you really talk about lizards in general of a guild?

Minor comments

L62: substitute "design" by "designed"

L83: change "the latter" by "lizards"

L161: I am not sure I would exclude those sites since you may overestimate predictor effects.

L166-167: you have four surveys on each plot. I suppose there was variation on specie´s counts over surveys, due to several aspects, including seasonality (which you ignored). I advise on modeling this variability on counts.

L168-169: what time did you search? Are all lizards diurnal/nocturnal? Any potential bias here?

L175: I would rather call these ‘environmental predictors’, since gradients may remind other things as well.

L183: change ‘.’ by ‘,’

L221-223: maybe it is just me, but the way it is written seems that you avoided correlated models, not predictors.

L223-226: what was the cut-off point?

L233: what do you mean by “corrected for few parameters”?

L234-235: what about the non-nested models? Why using the cut-off point if important model weight may be left out? Isn´t model averaging a better strategy depending on the case?

L261-262: how can you be sure those species were not there?

Table 2. what kind of models did you build? Only those with main effects, excluding additive and interactive effects? Did you try combining all best predictors in one model?

Figure 3. which model did you use to plot the relationship between abundance and number of trees? The top model or all three models? Notice that model weights are similar and all of them include the number of trees.

Table 3: This table is somewhat hard to read. Please, insert a backspace between West and East or a line between them. In addition, maybe you could reduce the names of the predictors or show their first letters only.

L293: again, I would rather call ‘predictor’ or ‘covariate’ instead of ‘gradient’.

L294: which model? Not the top model nor the third best-ranked model, if I read the table correctly (please, revise the table configuration). However, all models present similar AIC weights. How was the rationale behind using the only model that presented a significant effect? Did you try model averaging? I would not pick just one model among several to present a result, especially if they diverge.

Figure 4. Same comment as in Figure 3.

L352: there is a typo in this line.

L389-390: I am not convinced about that without including temporal predictors in the analysis. Was it too hot or too cold during some of the samplings? Were there thunderstorms or heavy rainfall in, at least, some of the sampling occasions?

L400-405: I find the discussion on conservation implications interesting, but I am not sure you provided all the elements for that. For instance, what kind of forest did you measure, native on invasive? Were there pine plantations (which are also categorized as forests)? How is the relationship between lizard assemblage and forests in general?

L406-407: please, see my major comment on this result, which seemed to be based not on the top-model in one analysis.

L419: you start this paragraph stating that species turnover was influenced by the distance from the river. But this was not found in all samples, as you stated in the same paragraph. Given that your samples covered only 5 km from the river, which in the Amazon, does not seem to be a huge area, what distance do you think you would find such effect?

L427-428: what do you mean by "...was returned..."?

L451-452: what do you mean by "efficient methods for multi-scale sampling"? Can you provide suggestions about what you would say is an efficient method?

6. PLOS authors have the option to publish the peer review history of their article (what does this mean?). If published, this will include your full peer review and any attached files.

Reviewer #1: No

---

## [Author Response · Author response to Decision Letter 0]

2 Mar 2020

Responses to editor comments:

Response: All PloS ONE style standards have been revised.

2. We note that Figure 1 in your submission contain [map/satellite] images which may be copyrighted. All PLOS content is published under the Creative Commons Attribution License (CC BY 4.0), which means that the manuscript, images, and Supporting Information files will be freely available online, and any third party is permitted to access, download, copy, distribute, and use these materials in any way, even commercially, with proper attribution. For these reasons, we cannot publish previously copyrighted maps or satellite images created using proprietary data, such as Google software (Google Maps, Street View, and Earth). For more information, see our copyright guidelines: http://journals.plos.org/plosone/s/licenses-and-copyright.

Response: The satellite image used to compose Figure 1 was changed following the proposed guidelines and a new image was added in its place from The Gateway to Astronaut Photography of Earth (public domain).

Response to Reviewers

1. My main concern is that all inference here is based on count and presence/absence data, which may be influenced by sampling errors. For instance, you may find 20 individuals of species A and 19 individuals of species B in a given plot, when in fact there was 120 individuals of species A and 20 individuals of species B. This is typically a false-negative type error. Alternatively, you may have misidentified species B, inducing false-positive errors in your counts. Similar problems occur when using presence/absence data when absence is not included and estimated in the model. This subject is well covered in the literature. You may want to have a look at Kéry and Schmidt 2008 (Community Ecology 9: 207-216), Fitzpatrick et al. 2009 (Ecological Applications 19: 1673-79), Miller et al. 2012 (Ecological Applications 22: 1665-74) and so on. Since sampling effort seems to be enough, I recommend applying a modeling technique that accounts for sampling errors (for instance, N-mixture models), which I find more appealing to discuss results arising from count data. However, I understand that you would have to make a major change in your manuscript, especially if results change, which I would not be surprised. At least, I recommend including a paragraph discussing this subject, since I find hard to believe that you have captured the exactly number of individuals and species present at a given site, and thus, the results may be biased.

Response:

We agree with the reviewer that when we study ecological data based on abundance and presence / absence data, we may be masking sampling errors since every technique is biased and will depend on the probability of species detection. However, as in our study we conducted multiple data collections at every point, combined with more than one sampling method, we think that we minimized sampling bias as much as possible.

As for the suggested modeling technique (N-mixture models): we opted for the second option suggested by the reviewer, which would be to continue using the models already proposed and not change our analyzes. We add a paragraph discussing these type I and II errors and the possible biased results generated from the sampling and that could be altering our results in some way. We justify our decision not to use N-mixture models because the same consist of two linked generalized linear models, however, for the realization of this model, the ordering method chosen by us (principal coordinate analysis - PCoA) cannot be used since it does not support the input of negative values. Given that the PCoA returns negative values of scores on its axes we could not be using this sorting method. Therefore, we chose to continue with PcoA to the detriment of the other sorting methods that could be substituted to meet the assumption of generalized linear models, since our choices followed the one suggested by GOTELLI & ELLISON (2011, page 446 of the book Principles of Statistics in Ecology). These authors recommend the use of PcoA for “sorting applications in which the objective is to preserve the original multivariate distances between observations in the reduced space (sorting)”, especially when we intend to use the axes generated in the ordering in later analyzes. With regard to the use of other methods such as the widely used NMDS, the axes are not orthogonal, that is, we could not choose just one of the axes as a predictor of our species composition.

We carried out several previous analyzes with other sorting methods, different types of returns, and simulations including the removal of species that presented low abundance. In each simulation we had the precaution to show to other researchers in the area and request their opinions before deciding on the methods presented in this version of our manuscript. Among these simulations, we carried out the NMDS which captured 46.64% of the variation in species composition with two axes and demonstrated that the lizard assemblages differed significantly in relation to the banks of the Madeira River (MANOVA: Pillai trace = 0,205, F1,80= 10,229, p <0,001). As for the results of the regressions with the environmental variables, the ordering of the NMDS plots for the left side of the river was only possible using three NMDS axes due to the little variation in the data with two axes (which led us to prefer the analysis main coordinates - PcoA), and as a result captured 70% of the variation in species composition. The explanation for the right margin using two NMDS axes was 57%. As for the possible structuring environmental variables, the effect of the vegetation structure (number of trees) on the left side (F7,38=2,567, p=0,04), and for the right margin, only elevation was identified as a factor influencing the composition of the lizard assemblages (F1,70=7,737, p<0,001). With all the simulations with our data we can see a strong effect of the modules themselves and the need to analyze the modules Ilha da Pedra (west bank) and Jirau-D (east bank) separately from the rest of the sample units, so we arrive at our current results .

2. Your sample includes temporal variation (extended for a year, or so) but there are no temporal covariates among your predictors. Lizard activity is usually related to temperature, sunlight, rainfall and so on. You probably sampled plots in different days and different hours within a day, and ciclicity occurs even within individuals. How do you think the absence of temporal predictors affect your results? For instance, Figure 2 shows species ‘abundances on each river bank. Some species seems to be common on one bank, but not the other. Is this really the barrier effect of the river? By ignoring temporal variability, you assume sampling did not vary along the year.

Response:

Each sampling campaign was carried out at a different time of the year: Campaign I- 24 February to 26 April 2010; Campaign II- July 30 to August 19 2010; Campaign III- November 5 to 26 2010; and Campaign IV- January 13 to February 4 2011. In each of these campaigns, all plots inserted in the different modules were sampled, that is, each plot was sampled in four different times of the year. Therefore, we think we minimized the sampling bias regarding the presence / absence of species in the plots. In fact, the activity of the lizards is closely related to variations in temperature, light and precipitation in a given environment. With this in mind, the plots were sampled at different times. We believe that our sampling method that combined active visual search, with two simultaneous observers, supplemented by sweeping the leaf-litter, minimized the lack of temporal predictors in the presence/absence results.

3. Authors suggest that the pattern found – the river acting as a barrier – was important. However, they mention that the lizard fauna found is widely distributed over the Amazon. This is somewhat confusing to me because if species were well-distributed, they would probably be found everywhere. To explain that, authors mention that the barrier effect of the river may vary, suggesting seasonality. This makes me wonder if species are truly isolated, as suggested. The barrier effect could be only temporarily and the results found are explained by the time frame of the study. I would like authors to clarify that point.

Response: 

Actually, most of the lizard species found in our study are widely distributed across Amazonia, and when we say that these species are isolated, we are not trying to extrapolate our results to the entire length of the river, so we were careful to keep that registered in the discussion (from line 385). as we do not have information for other locations along the length of the Madeira River. 

We did not want to suggest seasonality when we wrote that the river barrier effect may vary. The different Amazonian rivers (and even different parts of a same large river) have distinct widhts, current speeds, water types and geological dynamics, which may affect their role as an effective geological barrier.

Even widely distributed species in Amazonia are not necessarily found everywhere, since the configuration of the Amazon Basin can be related to a mosaic of different phyto-regions, observed both on a local and regional scale, with different geological ages and origins between the different fractions of the basin, a fact that leads to historical evolutionary differences between areas and that become determinant factors for the differences in lizard species richness in the region. Even within each area of endemism, with no current geographical barriers, ecological differentiation can occur (Ferrão et al., 2017; Ortiz et al. 2018). The isolation of lizard species in one of the banks of the river may be related to geomorphological heterogeneity along the course of the Madeira River, and even at different sampling dates, we still observe the restricted presence of some species to certain banks of the river.

Ferrão, M.; Moravec, J.; Fraga, R.; Almeida, A.P.; Kaefer, I.L.; Lima, A.P. 2017. A new species of Scinax from the Purus–Madeira interfluve, Brazilian Amazonia (Anura, Hylidae). ZooKeys, 706: 137–162.

Ortiz DA, Lima AP, Werneck FP (2018) Environmental transition zone and rivers shape intraspecific population structure and genetic diversity of an Amazonian rain forest tree frog. Evolutionary Ecology 32(4):359-378

4. I think your manuscript would benefit from contrasting your results with other lizard systems around the world. For instance, are there examples of rivers acing as barriers for lizards elsewhere? What about other reptiles? What are the covariates related to isolation for lizards or reptiles in general? In addition, you could provide some information about the Amazon basin. For instance, what is the mean width of the big rivers? Finally, concerning the species found. What is the habit of the most abundant species? Did you find terrestrial, arboreal and fossorial species equally? Does you results relate to species ‘habits? I believe that finding terrestrial species (the teiids, for instance) may be easier than arboreal (Enyalius?) or fossorial. In the same way, diurnal species will be easier to find during the day and the opposite for nocturnal. How was the sampling scheme? Can you really talk about lizards in general of a guild?

Response: 

 We thank the reviewer and agree that some information needs to be added, so we have included in the discussion other examples of lizards in the Amazon and around the world, and data on the covariates associated with restriction of geographic distributions. Information on the width of the largest Amazonian rivers was added to the methodology when mentioning the Madeira River, as it is one of the largest and most important rivers in the Amazon basin. 

Regarding the species habits, we found the tree species Gonatodes humeralis, and Norops fuscuoratus as the most abundant, followed by the teiid Ameiva ameiva. The fossorial or semi-fossorial species were the ones with the lowest abundances, a fact already expected and observed in different studies on lizard assemblages. We think that the results are related to the habits of the species, as included in the discussion (starting from line 428), where we associate the results of the gradient number of trees to some tree species found. 

Regarding the species activity time (more detail in the following answers to the reviewer below), most species are diurnal or crepuscular and few are nocturnal, but easily detectable to experienced researchers in the region.

 We observed that our sample is within the richness already highlighted in the local and regional literature for Amazonia, and we think we gave a good representation of how the lizard community in that region is structured. 

 With regard to the term guild, we decided not to apply because the observed species use different and complementary resources, thus not constituting a specific guild with regard to diet or habitat use, for example. 

Minor comments

✓ 

L62: substitute "design" by "designed" 

Response: Replaced in text

✓ 

L84: change "the latter" by "lizards"

Response: Replaced in text

✓ 

L161: I am not sure I would exclude those sites since you may overestimate predictor effects.

Response: We decided to maintain the exclusion, as some of these sites coincided with areas of difficult access, for example, due to fallen trunks, making it difficult to travel along the plot, or even partially flooded, and could contribute to the low probability of detection. Thus, we decided to exclude all sites that contained zero individuals.

✓ 

L166-167: you have four surveys on each plot. I suppose there was variation on specie´s counts over surveys, due to several aspects, including seasonality (which you ignored). I advise on modeling this variability on counts.

Response: Question previously answered.

✓ 

L168-169: what time did you search? Are all lizards diurnal/nocturnal? Any potential bias here?

Response: Most of Amazonian lizards are diurnal. Even those that start their activity at dusk, such as Uranoscodon superciliosus (Linnaeus, 758), are easy to detect because they usually rest over tree branches while inactive. Regarding campaign schedules, all were carried out during the day. However, the first and last were carried out in the afternoon and early evening, and the second and third were carried out in the morning and early afternoon. Thus, we contemplated different activity periods, maximizing detectability.

✓ 

L177: I would rather call these ‘environmental predictors’, since gradients may remind other things as well.

Response: Replaced in text.

✓ 

L183: change ‘.’ by ‘,’

Response: Replaced in text.

✓ 

L221-223: maybe it is just me, but the way it is written seems that you avoided correlated models, not predictors.

Response: The sentence was rewritten for clarity.

✓ 

L223-226: what was the cut-off point?

Response: From r = 0.51. We added this information in the text.

✓ L233: what do you mean by “corrected for few parameters”?

Response: We wanted to say that we did use Akaike´s Information Criterion (AIC) for model selection, but the one corrected for small samples: the AICc. The sentence was poorly written in the first version and rephrased for clarity.

✓ 

L234-235: what about the non-nested models? Why using the cut-off point if important model weight may be left out? Isn´t model averaging a better strategy depending on the case?

Response: I am not sure if I understood the question, and in what our analysis differs from what has already been done in the literature. We stipulated the ∆AICc <2 cut, due to the number of possible models with different combinations between the variables, and this cut would inevitably exclude some models at the expense of those more parsimonious models. 

✓ 

L261-262: how can you be sure those species were not there?

Table 2. what kind of models did you build? Only those with main effects, excluding additive and interactive effects? Did you try combining all best predictors in one model?

Figure 3. which model did you use to plot the relationship between abundance and number of trees? The top model or all three models? Notice that model weights are similar and all of them include the number of trees.

Table 3: This table is somewhat hard to read. Please, insert a backspace between West and East or a line between them. In addition, maybe you could reduce the names of the predictors or show their first letters only.

Response: Yes, several models were built, with different combinations between different predictors that did not correlate with each other. In relation to figure 3, we show a direct ordination of the values from the gradient number of trees and the relative abundance of species by plots. With regard to the table, we added additional lines as suggested by the reviewer for a better visualization.

✓ L293: again, I would rather call ‘predictor’ or ‘covariate’ instead of ‘gradient’.

Response: Replaced in text.

✓ 

L294: which model? Not the top model nor the third best-ranked model, if I read the table correctly (please, revise the table configuration). However, all models present similar AIC weights. How was the rationale behind using the only model that presented a significant effect? Did you try model averaging? I would not pick just one model among several to present a result, especially if they diverge.

Figure 4. Same comment as in Figure 3.

Response: We decided to graphically show only the significant models (P < 0.05). We could really include the images of all the models (which were marginally significant), but we wanted to be synthetic and illustrate these results with a single image due to the length of the article.

✓ 

L352: there is a typo in this line.

Response: Corrected in text.

✓ 

L389-390: I am not convinced about that without including temporal predictors in the analysis. Was it too hot or too cold during some of the samplings? Were there thunderstorms or heavy rainfall in, at least, some of the sampling occasions?

Response: Samples were not taken in the rain. When this happened the samplings were postponed.

✓ 

L400-405: I find the discussion on conservation implications interesting, but I am not sure you provided all the elements for that. For instance, what kind of forest did you measure, native on invasive? Were there pine plantations (which are also categorized as forests)? How is the relationship between lizard assemblage and forests in general?

Response: All sampling modules were installed in original, preserved Amazonian terra-firme forests that are not influenced by other types of vegetation.

✓ 

L419: you start this paragraph stating that species turnover was influenced by the distance from the river. But this was not found in all samples, as you stated in the same paragraph. Given that your samples covered only 5 km from the river, which in the Amazon, does not seem to be a huge area, what distance do you think you would find such effect?

Response: We believe that species turnover is related to several aspects in addition to the distance from the river, and that stipulating a margin distance value for the Amazon is a risk, since the region has a large environmental heterogeneity between the interfluvial regions, and even over small distances of 5km we could have significant results in the turnover of species, especially those less vagile. However, many gaps need to be filled for Amazonian lizards, despite recent efforts to answer such a question.

✓ 

L427-428: what do you mean by "...was returned..."?

Response: We used returned in the sense that it was observed, but the word was replaced in the text.

✓ 

L451-452: what do you mean by "efficient methods for multi-scale sampling"? Can you provide suggestions about what you would say is an efficient method?

Response: Methods that present a spatial standardization, easily replicable for different species (modular), and preferably that can be carried out for a long time. An example of a good method is the one used here in our study, the RAPELD- Rapid Assessments and Long-term Ecological Research (Magnusson et al., 2005). These modules are part of a network of permanent standardized transects installed in the Amazon by the Biodiversity Research Program (PPBio) of the Brazilian Science, Technology, Innovations and Communications Ministry (Magnusson et al., 2013). An example of sample desing is shown in the figure below. In this method the plots follow the terrain level curve, minimizing possible topographic (environmental) effects within the plot (sampling unit). RAPELD has already been used in several localities around the world https://ppbio.inpa.gov.br/sites/default/files/Biodiversidade_e_monitoramento_ambiental_integrado.pdf. cited 10 March 2019.

---

## [Decision Letter · Decision Letter 1]

27 Apr 2020

PONE-D-19-15180R1

Hierarchical effects of historical and environmental factors on lizard assemblages in the upper Madeira river, Amazonian Brazil

PLOS ONE

Dear Mrs. Marques Peixoto,

Thank you for submitting your manuscript to PLOS ONE. After careful consideration, we feel that it has merit but does not fully meet PLOS ONE’s publication criteria as it currently stands. Therefore, we invite you to submit a revised version of the manuscript that addresses the points raised during the review process.

There was one referee and this person and myself find that the authors have done a good job to improve their paper. There a few minor things pending only, see attachment.

We would appreciate receiving your revised manuscript by Jun 11 2020 11:59PM. To enhance the reproducibility of your results, we recommend that if applicable you deposit your laboratory protocols in protocols.io, where a protocol can be assigned its own identifier (DOI) such that it can be cited independently in the future. For instructions see: http://journals.plos.org/plosone/s/submission-guidelines#loc-laboratory-protocols

We look forward to receiving your revised manuscript.

Kind regards,

Stefan Lötters

Academic Editor

PLOS ONE

Reviewers' comments:

Reviewer's Responses to Questions

**Comments to the Author**

1. If the authors have adequately addressed your comments raised in a previous round of review and you feel that this manuscript is now acceptable for publication, you may indicate that here to bypass the “Comments to the Author” section, enter your conflict of interest statement in the “Confidential to Editor” section, and submit your "Accept" recommendation.

Reviewer #2: All comments have been addressed

2. Is the manuscript technically sound, and do the data support the conclusions?

Reviewer #2: Yes

3. Has the statistical analysis been performed appropriately and rigorously? 

Reviewer #2: Yes

4. Have the authors made all data underlying the findings in their manuscript fully available?

Reviewer #2: Yes

5. Is the manuscript presented in an intelligible fashion and written in standard English?

Reviewer #2: Yes

6. Review Comments to the Author

Reviewer #2: Most of the suggestions and corrections by a previous reviewer were accepted and incorporated by the authors. Some pending questions and additional suggestions, as well as general comments and editorial changes have been added to the present review.

7. PLOS authors have the option to publish the peer review history of their article (what does this mean?). If published, this will include your full peer review and any attached files.

Reviewer #2: No

---

## [Author Response · Author response to Decision Letter 1]

6 May 2020

Manaus, May 5th 2020.

RESPONSE LETTER

PONE-D-19-15180 - Second round of review 

Hierarchical effects of historical and environmental factors on lizard assemblages in the upper Madeira River, Brazilian Amazonia

Dear Dr. Stefan Lötters, 

Thank you very much for such detailed revisions on our manuscript. Please find below our point-by-point letter addressing all comments and suggestions. Responses are marked in blue.

Kind regards,

The authors

 Responses to comments:

Dear all, 

A previous reviewer expressed his/her concerns on possible sampling errors due to the methods used to estimate species presence/abundance. The authors answered that some of those errors were minimized by using “more than one sampling method”. Indeed, besides the visual searching, they only referred searching amidst leaf litter or under debris, and stated that this could enhance detection of fossorial/semi-fossorial species. However, there is no strictly fossorial species in the assemblage reported, and such species in reality could barely be found just by sweeping the leaf-litter. Moreover, although the authors provided responses to most of the reviewer’s concerns, they made no comment on the possibility that some of the taxa indeed represent cryptic species. I suggest them to discuss this it a little bit. 

Response: Thank you very much for these comments. In this revised version of the manuscript we highlighted that "search on the vegetation and on the leaf litter was systematically conducted in the first and second half of each visual sampling in the plots, respectively, thus constituting two different and complementary methods". With regard to the assemblage of fossorial species, now we acknowledge that it was not sampled in this study, thus restricting our sample universe and reinforcing confidence to our results. Now we state in the abstract and methods that "we use arboreal and leaf-litter lizard abundance data". 

Some of the points raised by the reviewer on item 4 of his/her review also deserve additional efforts from the authors. As an example, mean width of the upper Madeira River was not mentioned anywhere, but this is an important issue when hypothesizing it could act as a riverine barrier. Also, information on sampling scheme are not yet clear enough. Visual searches were performed during the day only, or also during the night, since there are nocturnal species in the assemblage studied? 

Response: We added the followed passages in order to attend to these pertinent suggestions: "...and its width varies from about 0,5 to 10 km depending on the river flow. The Madeira separates the Inambari and Rondônia endemism zones located along its left (west) and right (east) margins, respectively [20].". "The searching time in each plot varied between 40 and 60 minutes and was always conducted during the day.". We are not restricting the assemblage to diurnal lizards in the manuscript because species that are usually active during the night were also sampled in daytime hours. 

The former reviewer also pointed that “authors suggest that the pattern found – the river acting as a barrier – was important. However, they mention that the lizard fauna found is widely distributed over the Amazon.” He then made some comments on this topic. The authors provided a pertinent response but I think the discussion on this topic could benefit from adding supporting literature on known distributions of some of the species treated therein, which are based in more historical datasets and which clearly show the pattern discussed. 

Response: In Discussion, we clearly state that "We highlight that most of regionally isolated species in our sample are widely distributed throughout Amazonia outside our study area [13]...". In addition, in this revised version we acknowledge in the Abstract that "This finding suggests species have been historically isolated at one of the river banks, or that distinct geomorphological features influence species occurrence at each river bank.". 

Minor comments Page 1 

Line 3 - "Brasil Amazônico" does not make sense... Brazilian Amazonia would be more appropriated 

Response: Replaced by "Brazilian Amazonia" throughout the text.

Page 2 

Line 36 - you have used "river" in the title and "River" from now on. Check and standardize it throughout the text - River or river. 

Response: Replaced by "Madeira River" throughout the manuscript.

Line 39 - delete space 

Response: Space suppressed. 

Line 44 - "along" would be better 

Response: Replaced by "along" in three phrases. 

Line 46 - "... over the extensive (nearly xxx km) study area." (I suggest you to briefly mention in the abstract the magnitude of the area to which these results aplly) 

Response: Replaced by "over the extensive (nearly 500 km) study area. ". 

Page 3 

Line 35 - Brazilian Amazonia 

Response: Replaced by "Brazilian Amazonia". 

Line 72 - first cite frogs, then birds. Please also sort citations in ascending order, throughout the text. 

Response: Replaced by "...explaining limited distribution of plants, frogs, birds, spiny rats, and primates". Citations were sorted in ascending order in this new version of the manuscript. 

Line 72 - first cite spiny rats, then primates 

Response: Replaced by "...explaining limited distribution of plants, frogs, birds, spiny rats, and primates".

Page 4 

Line 82 - insert an Oxford comma before "and"; see additional remarks on it in line 182 

Response: Comma added. 

Line 82 - Ideally, you should cite them in an evolutionary rank. Then you will need to renumber references. Please also sort citations in ascending order. 

Response: Now cited in evolutionary rank.

Line 84 - sort citations in ascending order 

Response: Citations were sorted in ascending order in this new version of the manuscript. 

Line 85 - sort citations in ascending order 

Response: Citations were sorted in ascending order in this new version of the manuscript. 

Page 5 

Line 116 - basin (the "megadiverse region" you want to better understand is that one under the influence of the river, not the river itself). 

Response: We added "basin".

Line 120 - delete space 

Response: Space deleted.

Line 120 - insert space 

Response: Space inserted.

Line 122 - As one of your hypothesis treats the upper Madeira as a barrier, please add some information on relevant characteristics of the river itself (e.g., river width). 

Response: We added the following passage: "and its width varies from about 0.5 to 10 km depending on the river flow.".

Line 123 - Brazilian

Response: "Brazilian" inserted.  Line 123 - River (standardize) 

Response: Standardized in uppercase letter.

Line 124 - River 

Response: Changed to "River". 

Line 124 - check standards of the journal (Fig instead of Fig.). Additionally, please add some information on the endemism zones (Rondônia and Inambari), so the mention to them can be properly evaluated when analyzing the elements in your Fig 1. 

Response: Standardized to "Fig" instead of "Fig.". We added the following passage: "The Madeira separates the Inambari and Rondônia endemism zones located along its left (west) and right (east) margins, respectively [20].". 

Page 6 

Line 126 - And what does it mean? These two endemism zones are only briefly mentioned in the Introduction and here in the legend for Figure 1. You should briefly discuss on the implications of this in your results. 

Response: We added the following passage to Discussion: "In the upper Madeira River, assemblage divergence between river banks has been attributed to historical processes regionally reducing species dispersal [12], and delimiting the Amazonian endemism zones Inambari and Rondônia [20].". 

Page 7 

Line 153 - check and standardize throughout the text: you used either with or without hyphen 

Response: Standardized as East and West Jirau (without hyphen). Jaci-Paraná has a hyphen because this is the original name of the locality. 

Line 154 - see comment on line 153 

Response: Standardized as East and West Jirau (without hyphen). Jaci-Paraná has a hyphen because this is the original name of the locality.

Line 103 – replace “e” by “and” 

Response: Replaced by "and". 

Line 106 - Brazilian Amazonia

Response: Replaced by "Brazilian Amazonia". 

Lines 165-166 - Dates should follow a standardized sequence (e.g.,day-month- year, as you used here) 

Response: Replaced by "We sampled each plot in four different periods (24 February to 26 April 2010, 30 July to 19 August 2010, 5 November to 26 November 2010, and 13 January to 4 February 2011)". 

Line 172 – there is no fossorial species in your list. 

Response: "fossorial" was excluded. 

Line 174 - at what time of the day/night? 

Response: Altered to "The searching time in each plot varied between 40 and 60 minutes and was always conducted during the day.". 

Page 8 Line 178 – replace squamates by lizards '

Response: Changed to "lizard". 

Line 181 – soil in lowercase 

Response: "soil" now in lowercase. 

Line 182 - Note: check journal style and consistently use (or do not use) the Oxford comma throughout all the text. In this paragraph, as an example, you alternatively used it (see line 180 - "soil texture, fertility, and flat-level") and did not use it (e.g., in lines 179 - "...foraging, resting and thermoregulation sites", and line 182 - "...Potassium, Zinc and exchangeable Aluminum"). 

Response: We opted to use the Oxford comma throughout the manuscript. The entire manuscript was revised with regard to the commas. 

Line 184 – remove the dot before citation [61] 

Response: Dot removed. 

Page 9 Line 204 - along the same river bank (replace "within a" by "along the same") 

Response: Changed to "...modules along the same river bank...".

Page 10 

Line 224 - insert space between "predictors" and "that" 

Response: space inserted. 

Line 247 – bring to the suggested point the mention to Table 1, or, alternatively, to a point immediately after the mention of the three most abundant species. The information on the number of plots where each species was found do not appear in the table. 

Response: We brought the mention of Table 1 to the point immediately after the three most abundant species.

Page 11 

Line 251 - remove the mention to Table 1 here, and insert it at any point in the previous sentence. There is no information in Table 1 on the number of plots where each species was found. 

Response: Mention removed. 

Table 1 – check correspondence of values to each species: the values highlighted possibly correspond to Alopoglossus angulatus. Please relocate them to the correct line of the table. 

Response: Values relocated to Alopoglossus angulatus.

Page 12 Table 1 – complete year of description of Kentropyx calcarata and of 

Uranoscodon superciliosus

Response: Values completed: 1825 and 1758, respectively. 

 Line 265 - insert an Oxford comma after bassleri

Response: Comma added.

 Line 267 - "and" not in italics. Please also add an Oxford comma before "and" . 

Response: Replaced by "Kentropyx calcarata, and Enyalius leechii...".

Line 272 - replace "of" by "in" ; replace “and” by “or” 

Response: Changed to "Plots ordinated according to their position in the upper Madeira River (west or east bank).". 

Line 273 – add “bank” after “east” 

Response: Changed to "Plots ordinated according to their position in the upper Madeira River (west or east bank).".

Page 13 

Line 286 - insert an Oxford comma after “Búfalos” 

Response: Oxford comma added. 

Line 288 - standardize mention to the Akaike's criterion: "?AICc < 2", as used before. 

Response: Standardized to "ΔAICc < 2". 

Page 14 

Table 2 - correct the word "margim". Check it in every Table. It appears more than once. 

Response: Word corrected to "margin" throughout the table. 

Line 292 – insert "to" before "the number" 

Response: Corrected to "position relative to the number of trees...". 

Page 15 

Line 305 - standardize mention to the Akaike's criterion: "?AICc < 2", as used before. 

Response: Standardized. 

Table 3 - correct the word "margim". Check it in the table. 

Response: corrected to "margin". 

Page 16 

Table 3 - correct the word "margim". 

Response: Word corrected. 

Lines 312-313 – Principal Coordinates Analysis (cite it according to the abbreviation PCA) 

Response: Replaced by "Principal Coordinates Analysis".

Page 17 Line 331 - “East Jirau” without hyphen. Standardize it throughout the text. 

Response: East Jirau was standardized without hyphen. 

Line 332 - insert an Oxford comma between "sand, and clay"

Response: Comma inserted. 

 Line 333 - see comments on line 312.

Response: Changed to "Principal Coordinates Analysis". 

 Line 342 - ...[18], diurnal frogs [12], birds... (cite diurnal frogs before birds) 

Response: "diurnal frogs" now cited before "birds". 

Line 346 – “...with those obtained for frog assemblages” 

Response: Replaced by "Our overall results are broadly consistent with those obtained for frog assemblages sampled in the same plots...". 

Page 18 

Lines 355-356 or elsewhere: did you ever discuss on the possibility of taxonomic bias contributing to this scenario? There are consistent and recent efforts to investigate the taxonomic status of frog species in the region, and such efforts are unparalleled regarding lizards. Maybe you can use recent literature to briefly discuss on that. Only 19% of the species in Table 1 were described in the 20th century, while 31% of the frogs considered in Dias-Terceiro et al. (2015) were described in 21th century and other three species were recognized as undescribed by the same authors. 

Response: With regard to this pertinent comment, in the revised version we state that "The smaller proportion of regionally isolated lizard species (29.63%) is reasonably explained by the lower dispersal capacity of small and site-attached frogs compared with most lizards. A taxonomic bias may be also contributing to this scenario, since there are consistent and recent efforts to investigate the taxonomic status of frog species in the region [12], and such efforts are unparalleled regarding lizards.". 

Line 360 - you could mention the distribution patterns for some of the species considered herein, depicted in Ribeiro Jr. (2015), as an additional source of evidence that the pattern you found is real, even though a few specimens were recorded during your study (the distribution shown for E. leechii, as an example, based in a large dataset). 

Response: In Discussion, now we clearly state that "Even though the riverine-restricted geographic distribution of species such as Kentropyx calcarata observed in this study is supported by basin-level data, we highlight that most of regionally isolated species in our sample are widely distributed throughout Amazonia outside our study area [13].". In addition, in this revised version we acknowledge in the Abstract that "This finding suggests species have been historically isolated at one of the river banks, or that distinct geomorphological features influence species occurrence at each river bank.". 

Lines 362-363 - again, taxonomic bias may be also one of causes of such inconsistencies. What you are calling Kentropyx altamazonica, as an example, potentially represents a cryptic species, as other taxa in your Table 1. This is out of the scope of your work, but cannot be ignored 

Response: In the revised version we wrote that "The smaller proportion of regionally isolated lizard species (29.63%) is reasonably explained by the lower dispersal capacity of small and site-attached frogs compared with most lizards. A taxonomic bias may be also contributing to this scenario, since there are consistent and recent efforts to investigate the taxonomic status of frog species in the region [12], and such efforts are unparalleled regarding lizards.". 

Line 363 - sort citations in ascending order 

Response: sorted in ascending order. 

Page 19 

Line 385 - you basically used "active visual search". Did you additionally use a systematic search using leaf litter plots? Or you merely searched leaf litter anurans instead, by "sweeping the leaf litter and removing debris"? Looking carefully at every microhabitat visually accessible and inspecting potencial retreats are common procedures during visual searches. 

Response: Now we clearly state in Discussion: "However, we think that a possible sampling bias was mitigated by the large sampling effort associated to the combination of different visual sampling methods employed in this study.". We rephrased Methods for clarity and reached the followed wording: "We found lizards using active visual search, with two simultaneous observers positioned 10 m apart. In addition, we supplemented the sampling effort by sweeping the leaf litter and removing debris in a 2 m strip following the center line of the plot. This approach was particularly useful to increase the efficiency of sampling leaf-litter species (e.g. Alopoglossidae, Gymnophthalmidae). Search on the vegetation and on the leaf litter was systematically and respectively conducted in the first and second half of each visual sampling in the plots, thus constituting two different and complementary methods. The searching time in each plot varied between 40 and 60 minutes and was always conducted during the day.".

Line 401 - cite in an evolutionary rank

Response: Cited in evolutionary rank: "frogs [12], snakes [79], and bats [80].". References were reordered in the list. 

 Line 403 – replace "within a" by "along a same" 

Response: Replaced by "along a same riverbank". 

Page 20 

Line 410 - of which zoological group? is this a general trend for chordates? for reptiles? for tropical reptiles? your supporting references herein (58-60) are for snakes, including a temperate species. Can you replace or complement them with any citation on the effects of vegetation heterogeneity on tropical lizards? 

Response: Changed to: "Heterogeneity in vegetation structure affects occurrence and abundance of tropical squamates due to variation in the availability of foraging, nesting, resting, and thermoregulating sites [58,60].". We changed "species" by "tropical squamates", and replaced the reference from a temperate environment [60] by a tropical one. We used references on snakes because we did not find examples related to lizards - probably due to the relative few studies on tropical lizards. 

Line 413 - insert an Oxford comma after “humidity” 

Response: Comma added after "humidity". 

Page 21 Line 435 - insert an Oxford comma after “cover” 

Response: Comma added after "cover".

Line 437 – “...snake [41], and bird [39] assemblages.” 

Response: reordered in an evolutionary rank. 

Line 447: insert “upper” before Madeira River 

Response: changed to "both banks of the upper Madeira River...". 

Page 22 Line 463 - sort citations in ascending order 

Response: In the megadiverse Amazonian rainforests this has been achieved by a few studies [12,31,73].

Page 33 Line 725 - italicize Liolaemus monticola 

Response: "Liolaemus monticola" italicized. 

Page 34 Line 746 - not in italics 

Response: Italics removed from "Boulenger". 

Page 36 Line 803 - West Jirau (standardize)

Response: Standardized to "West Jirau".  Line 801 - check

Response: Changed to "Effect of environmental predictors on lizard assemblages composition".

Line 805 - check

Response: "Effect of environmental predictors on lizard assemblages composition".  Line 806 - East Jirau (standardize)

Response: Standardized to "East Jirau".   Line 806 – delete extra space before “Paraná” 

Response: Extra space deleted.

---

## [Editor Report · Decision Letter 2]

15 May 2020

Hierarchical effects of historical and environmental factors on lizard assemblages in the upper Madeira River, Brazilian Amazonia

PONE-D-19-15180R2

Dear Dr. Marques Peixoto,

We are pleased to inform you that your manuscript has been judged scientifically suitable for publication and will be formally accepted for publication once it complies with all outstanding technical requirements.

With kind regards,

Stefan Lötters

Academic Editor

PLOS ONE
---

## [Editor Report · Acceptance letter]

21 May 2020

PONE-D-19-15180R2 

Hierarchical effects of historical and environmental factors on lizard assemblages in the upper Madeira River, Brazilian Amazonia 

Dear Dr. Marques Peixoto:

I am pleased to inform you that your manuscript has been deemed suitable for publication in PLOS ONE. Congratulations! Your manuscript is now with our production department. 

With kind regards,

on behalf of

Prof. Dr. Stefan Lötters 

Academic Editor

PLOS ONE